# Effect of population inflow and outflow between rural and urban areas on regional antimicrobial use surveillance

**Ryuji Koizumi**[1], **Yoshiki Kusama**[1,2]*, **Yuichi Muraki**[3], **Masahiro Ishikane**[1,4], **Daisuke Yamasaki**[5], **Masaki Tanabe**[5], **Norio Ohmagari**[1,2,4]

**1** AMR Clinical Reference Center, National Center for Global Health and Medicine, Tokyo, Japan, **2** Department of Emerging and Re-emerging Infectious Diseases, Tohoku University School of Medicine, Miyagi, Japan, **3** Department of Clinical Pharmacoepidemiology, Kyoto Pharmaceutical University, Kyoto, Japan, **4** Disease Control and Prevention Center, National Center for Global Health and Medicine, Tokyo, Japan, **5** Department of Infection Control and Prevention, Mie University Hospital, Mie, Japan

* stone.bagle@gmail.com

## Abstract

**Data Availability Statement:** All relevant data are within the manuscript and its Supporting information files.

### Purpose

Regional-level measures can complement national antimicrobial stewardship programs. In Japan, sub-prefectural regions called secondary medical areas (SMAs) provide general inpatient care within their borders, and regional antimicrobial stewardship measures are frequently implemented at this level. There is therefore a need to conduct antimicrobial use (AMU) surveillance at this level to ascertain antimicrobial consumption. However, AMU estimates are generally standardized to residence-based nighttime populations, which do not account for population mobility across regional borders. We examined the impact of population in/outflow on SMA-level AMU estimates by comparing the differences between standardization using daytime and nighttime populations.

### Methods

We obtained AMU information from the National Database of Health Insurance Claims and Specific Health Checkups of Japan. AMU was quantified at the prefectural and SMA levels using the number of defined daily doses (DDDs) divided by (a) 1,000 nighttime population per day or (b) 1,000 daytime population per day. We identified and characterized the discrepancies between the two types of estimates at the prefectural and SMA levels.

### Results

The national AMU was 17.21 DDDs per 1,000 population per day. The mean (95% confidence interval) prefectural-level DDDs per 1,000 nighttime and daytime population per day were 17.27 (14.10, 20.44) and 17.41 (14.30, 20.53), respectively. The mean (95% confidence interval) SMA-level DDDs per 1,000 nighttime and daytime population per day were 16.12 (9.84, 22.41) and 16.41 (10.57, 22.26), respectively. The nighttime population-standardized estimates were generally higher than the daytime population-standardized

**Funding:** This work was supported by the Ministry of Health, Labor and Welfare (MHLW) research grant of Japan (20HA2003).

**Competing interests:** The authors have declared that no competing interests exist.

estimates in urban areas, but lower in the adjacent suburbs. Large differences were observed in the main metropolitan hubs in eastern and western Japan.

## Conclusion

Regional-level AMU estimates, especially of smaller regions such as SMAs, are susceptible to the use of different populations for standardization. This finding indicates that AMU standardization based on population values is not suitable for AMU estimates in small regions.

## Introduction

Antimicrobial resistance is a major global health concern, and there is an urgent need to reduce inappropriate antimicrobial use (AMU) as a countermeasure [1]. In 2016, the Japanese government published the National Action Plan on Antimicrobial Resistance, which highlighted the integral role that regional cooperation plays in complementing national anti-microbial stewardship programs [2]. In order to improve AMU within a specific region, it is first necessary to ascertain the region's actual drug consumption patterns. However, there is a lack of information on best practice methodologies for regional-level AMU surveillance.

Under Japan's universal healthcare system, patients are free to seek care at any medical institution across regional borders without restriction. Furthermore, regions can experience daily fluxes in population size due to the inflow and outflow of people across borders for work and schooling. Attempts to characterize AMU at the regional level may therefore be hindered by differences between the locations where antimicrobials are prescribed and where patients reside. Accordingly, there are fundamental difficulties in accurately ascertaining the actual AMU of a region. In regions characterized by construction and air pollution, population migration has been reported to affect urban development and exposure to pollutants [3, 4]. We also posit that the impact of population inflow and outflow on AMU estimates would increase as the size of the target regional unit decreases, but the extent of such an effect has yet to be explored.

To date, AMU in Japan has been examined at the prefectural level [5], but there is a lack of information on sub-prefectural regions. Because healthcare policies are frequently imple-mented at the sub-prefectural level, understanding the trends in AMU at this level can help to identify region-specific problems and support the development of more precise and effective antimicrobial stewardship programs.

National-level AMU is generally indicated using the number of defined daily doses (DDDs) per 1,000 inhabitants per day (DID) based on population statistics. Regional-level DID esti-mates are dependent on the definition of each region's population, which in turn is affected by population mobility across borders. However, the effects of different population definitions on DID estimates at the regional level are unknown. To improve our understanding of regional-level AMU surveillance methodologies, this study aimed to elucidate the impact of population inflow and outflow on sub-prefectural DID estimates in Japan.

## Materials and methods

### Japan's health insurance system and secondary medical areas

In Japan, all residents are required to enroll in health insurance, which enables them to receive healthcare at any medical institution throughout the country. Each enrollee's insurance plan is

dependent on his/her age and occupation. Enrollees pay monthly premiums to their insurers, and also pay a portion of the medical charges (i.e., copayments) at the point of care when receiving insurance-covered treatments and medications. These copayments range from 10% to 30% depending on the enrollee's insurance plan and income level. The healthcare providers send insurance claims to the applicable insurers through a claims processing agency in order to be reimbursed for the remaining charges.

Japan's healthcare provision infrastructure involves three increasing levels of geographical divisions—designated primary, secondary, and tertiary medical areas—that serve as units for the implementation of healthcare policies. Primary medical areas comprise the nation's municipalities, and are equipped to provide basic primary outpatient care. Secondary medical areas (SMAs) are sub-prefectural regions comprising several primary medical areas, and are designed to meet each region's need for general inpatient care (including emergency care). Tertiary medical areas are mostly represented by prefectures, and provide specialized care that requires advanced technology and equipment. Although primary and tertiary medical areas generally use existing regional borders, SMAs are delineated based on the presence of health-care facilities that enable them to fulfill their designated functions. SMAs, which are most frequently used as the basic unit for healthcare planning, include designated core hospitals that treat critically ill inpatients, specialized outpatient clinics, and hospitals that provide routine in-hospital treatment. As of December 2020, there are 344 SMAs located throughout Japan (see S1 Table for the list of SMAs and their constituent municipalities).

## Data source

For this retrospective study, data were obtained from the National Database of Health Insurance Claims and Specific Health Checkups of Japan (NDB). The NDB has collected and maintained insurance claims data provided by the Ministry of Health, Labour and Welfare since April 2009, and these data can be used for research purposes through the submission and approval of an application [6]. Because insurance-covered care accounts for the majority of medical treatments provided in Japan, the NDB represents a near-comprehensive database of all treatments performed throughout the country. However, the database does not include claims data from patients with fully publicly funded healthcare (e.g., patients with intractable diseases, atomic bomb survivors, patients on welfare, patients with tuberculosis, and patients with human immunodeficiency virus infections) and patients who personally pay for all of their medical expenses (e.g., foreign travelers and cosmetic surgery patients). In this study, we calculated the number of prescriptions generated for systemic antimicrobial drugs (both oral and injection) in each SMA in 2015. We also identified the SMAs where each prescribing healthcare facility and dispensing pharmacy were located. The NDB data were accessed in January 2020.

## Data processing

Antimicrobial drugs were identified using the J01 classification in the Anatomical Therapeutic Chemical/Defined Daily Doses system established by the World Health Organization's Collaborating Centre for Drug Statistics Methodology [7]. The populations used for analyses were the national population stratified by municipality (i.e., cities, towns, villages, and wards) and the daytime population published by the Statistics Bureau of the Ministry of Internal Affairs and Communications [8]. The national population estimate is the estimated size of the population on every October 1st that reflects the natural population growth, social dynamics, and nationwide migration of Japanese nationals. These parameters are reported for each year (from October 1st of the previous year to September 30th of the index year) based on

population data obtained from a national census of all households (conducted every five years) and the annual population inflow and outflow estimates. These population estimates have conventionally been used to calculate the national DID for AMU surveillance. As these estimates are based on residences, they represent the nighttime population. The daytime and nighttime populations at the national, prefectural, and SMA levels are presented in S2 Table.

The daytime population of a region accounts for the number of people at work or school, and was calculated based on the national census using the formula shown below.

*Daytime population = Nighttime population − (Outflow population + Inflow population)*

The nighttime and daytime populations of each SMA were calculated by totaling the respective populations of its municipalities.

## Analysis

The study period was 2015, which was the most recent year with available statistics on the daytime population. For this study, AMU was quantified using DID estimates. National-level, prefectural-level, and SMA-level DDDs were shown in S3 Table. We calculated and compared the national-level, prefectural-level, and SMA-level DID values that were standardized to either the nighttime population or daytime population. To calculate the AMU at the various levels, the DID values of their constituent regions were totaled. Next, we evaluated the distributions of daytime and nighttime population–standardized DID values at the prefectural and SMA levels using the Kolmogorov–Smirnov test. The mean DID values were compared with the national AMU using the one-sample *t*-test. The population-standardized DID at the prefectural and SMA levels were used to generate violin plots, which were examined to identify regions with notable discrepancies between the two types of populations. Correlations between nighttime and daytime population–standardized DID values were examined using Pearson's correlation coefficients.

We then calculated the difference between the nighttime population–standardized DID values and the daytime population–standardized DID values for each prefecture and SMA. Choropleth maps were created based on these differences, and the distributions of regions with substantial differences were examined. The Tokyo/Ku-chuoubu SMA was excluded from the choropleth map as it was an extreme outlier. Furthermore, we identified the SMAs with the largest positive and negative differences in DID values, as well as the SMAs with the smallest absolute differences.

Finally, we used the population number for each age category (children: <15 years, working-age adults: 15–64 years, and older adults: ≥65 years) for each SMA, and analyzed the correlation between the absolute difference in DID values and each age category. For this analysis, the Tokyo/Ku-chuoubu SMA was excluded as it was an extreme outlier.

## Data management

The mapping of prefectures was performed using Tableau version 2019.1.0 (Tableau Software, Washington, USA). For the visualization of the SMAs, geocoding was performed using the geographical information in Tableau based on the National Land Numerical Information published by the National Spatial Planning and Regional Policy Bureau of the Ministry of Land, Infrastructure, Transport and Tourism. Finally, correlations between the absolute difference in DID values in each SMA and the population age categories were examined using Pearson's correlation coefficients. Statistical analyses were performed using R ver 4.0.0 (R Core Team, Vienna, Austria), and *P* values below 0.05 were considered statistically significant.

### Ethics

The study did not involve any interventions in human subjects, and the NDB data were anonymized before being received by the authors. This study was approved by the institutional review board of the National Center for Global Health and Medicine (Approval Number: NCGM-G-002505-00).

## Results

Fig 1 shows the national-level, prefectural-level, and SMA-level DID values that were standardized to either the nighttime population or daytime population. When standardized with the nighttime population, the mean DID was 17.27 (95% confidence interval [14.10, 20.44]) at the prefectural level and 16.12 (95% confidence interval [9.84, 22.41]) at the SMA level. When standardized with the daytime population, the mean DID was 17.41 (95% confidence interval [14.30, 20.53]) at the prefectural level and 16.41 (95% confidence interval [10.57, 22.26]) at the SMA level. Both the prefectural-level and SMA-level DID values were normally distributed regardless of the population used for standardization. The daytime and nighttime population–standardized mean DID values at the prefectural level were not significantly different from the national-level DID values (nighttime: $P = .385$; daytime: $P = .811$); however, the corresponding mean DID values at the SMA level were significantly different from the national-level DID values (nighttime: $P < .001$; daytime: $P < .001$). As shown in Fig 1, daytime population–standardized DID at the SMA level had a narrower dispersion and a median value (center of the widest section in the violin plot) that was closer to the national-level DID than the nighttime population–standardized DID. In contrast, the daytime population and nighttime population–standardized DID at the prefectural level exhibited similar shapes in the violin plot.

Fig 2 shows scatter plots of nighttime population–standardized DID values against daytime population–standardized DID values. The correlation coefficient between nighttime and daytime population–standardized DID values was higher at the prefectural level (0.90; $P < .001$) than at the SMA level (0.80; $P < .001$). At the prefectural level, the nighttime population–

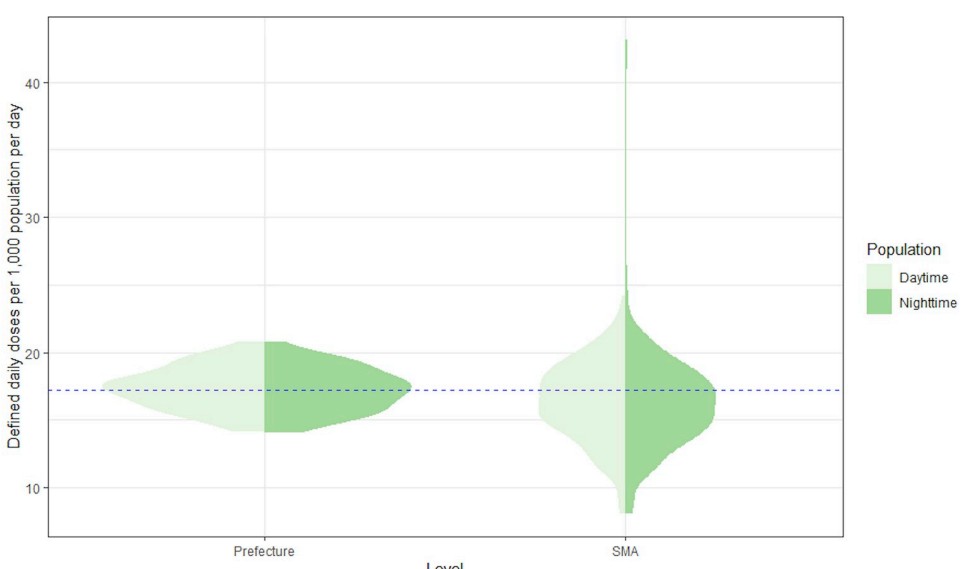

**Fig 1. Violin plots of prefectural-level and SMA-level DDDs per 1,000 nighttime population per day and DDDs per 1,000 daytime population per day.** The blue broken line represents the national-level antimicrobial use. DDD, defined daily dose; SMA, secondary medical area.

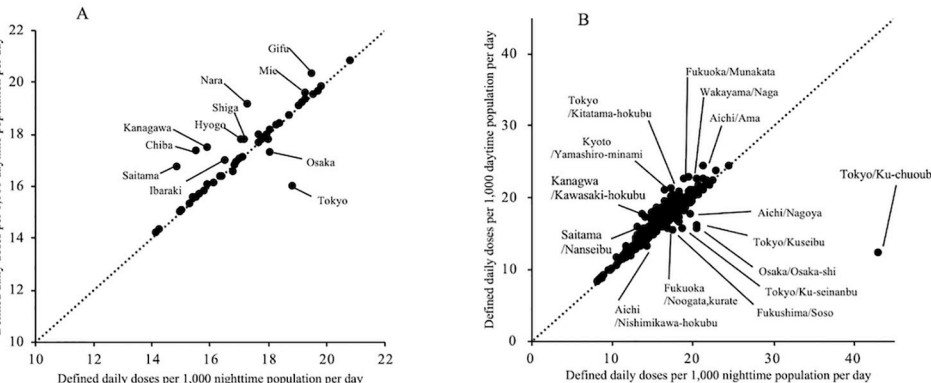

**Fig 2. Scatter plots of DDDs per 1,000 daytime population per day vs. DDDs per 1,000 nighttime population per day.** (A) Prefectural level and (B) Secondary medical area level. The dotted lines represent y = x. DDD, defined daily dose.

standardized DID values were higher than the daytime population–standardized DID values in central urban areas such as Tokyo and Osaka, but lower in surrounding prefectures such as Saitama, Kanagawa, Chiba, Gifu, and Mie. At the SMA level, the nighttime population–standardized DID value for Tokyo/Ku-chuoubu—which has a high concentration of companies and schools—was extremely high. In addition, the nighttime population–standardized DID values were higher in the urban SMAs of Tokyo/Ku-seibu and Osaka/Osaka-shi, but lower in surrounding SMAs such as Aichi/Ama, Wakayama/Naga, Fukuoka/Munakata, Tokyo/Kitatama-hokubu, and Kyoto/Yamashiro-minami.

Fig 3 shows choropleth maps of the differences between the nighttime population–standardized and daytime population–standardized DID values at the prefectural and SMA levels.

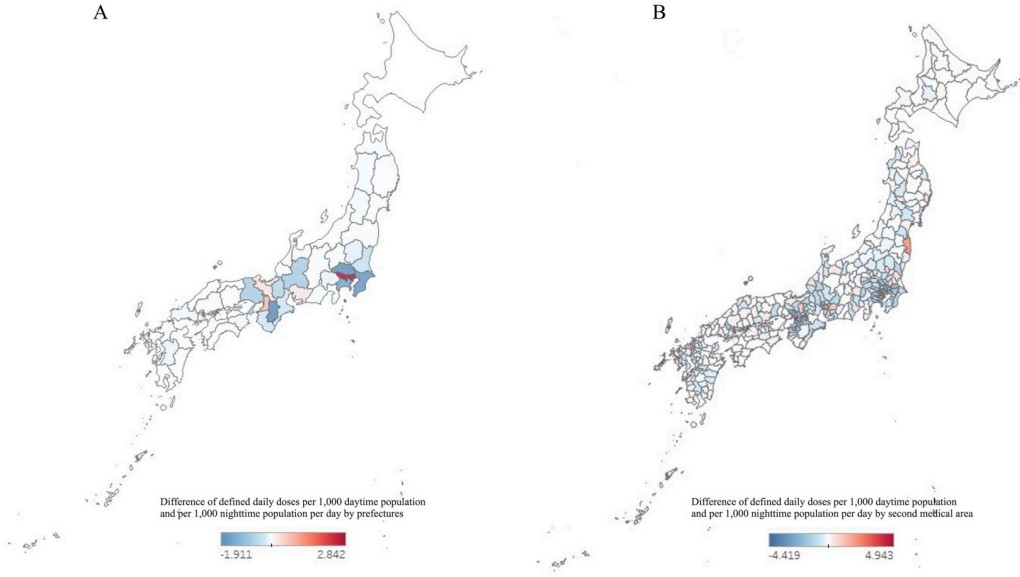

**Fig 3. Choropleth maps of Japan showing the differences between DDDs per 1,000 nighttime population per day and DDDs per 1,000 daytime population per day.** (A) Prefectural level and (B) Secondary medical area level. The numbers represent the differences between DDDs per 1,000 nighttime population per day and DDDs per 1,000 daytime population per day. The red and blue colors represent positive and negative differences, respectively. Data from Tokyo/Ku-chuoubu are not shown in Fig 3B because it was an extreme outlier (43.08). DDD, defined daily dose.

Among the prefectures, a positive difference (indicating that the nighttime population–standardized DID values were higher than the daytime population–standardized DID values) was observed for prefectures that contained more urban areas, such as Tokyo, Osaka, Aichi, and Kyoto. In contrast, a negative difference (indicating that the nighttime population–standardized DID values were lower than the daytime population–standardized DID values) was observed for prefectures adjacent to the urban areas. Prefectures showing substantial differences in DID values were limited to the eastern Kanto (which includes the Greater Tokyo Area) and western Kansai (which includes Osaka, Hyogo, Kyoto, and surrounding prefectures) metropolitan regions, with no notable differences observed in the other regions. Similarly, a positive difference in DID values was generally observed in urban SMAs, whereas a negative difference was observed in the surrounding SMAs.

The SMAs with the largest positive, largest negative, and smallest absolute differences between the nighttime population–standardized and daytime population–standardized DID values are presented in Table 1. Highly urbanized SMAs in Tokyo, Osaka, Aichi, and Fukuoka prefectures showed large positive differences. In contrast, SMAs in Kyoto, Tokyo, Fukuoka, Kanagawa, and Saitama prefectures (which are adjacent to urbanized areas) showed large negative differences. SMAs with the smallest absolute values were mostly found in rural prefectures such as Hokkaido, Shimane, and Kagoshima.

The nighttime population–standardized and daytime population–standardized DID values according to age group were calculated. Fig 4 shows the correlations between the absolute difference in DID values and the proportion of each age group in the population. The correlation coefficients were 0.14 ($P = 0.0082$) for children, 0.49 ($P < 0.001$) for working-age adults, and -0.44 ($P < 0.001$) for older adults.

## Discussion

In this retrospective nationwide study, we comparatively examined the effects of using the nighttime population and daytime population to adjust regional-level AMU in Japan. Previous studies have examined the effects of using different denominator values when calculating AMU in hospitals [9–12]. However, these effects on regional AMU surveillance have not been explored. Even the World Health Organization's AMU surveillance methodology does not address the appropriate methods for analyzing sub-national regions [13]. In our analysis, SMA-level AMU (standardized using either the daytime or nighttime population) was found to be significantly different from the national-level AMU. This suggests that the population-standardized DID values of smaller regions are susceptible to the effects of population inflow and outflow, which can lead to erroneous results.

As more medical examinations and prescriptions are received during the day than at night, the calculation of regional AMU indices should account for the effects of population inflow and outflow. At the SMA level, daytime population–standardization produced fewer outliers, narrower 95% confidence intervals, and mean DID values that were closer to the national DID than nighttime population–standardization. These findings showed that when analyzing smaller regional units such as SMAs, the use of different populations for standardization has a considerable effect on DID estimates in central urban areas and their surrounding regions. Our insights indicate that AMU standardization based on population values is not suitable for AMU estimates in small regions.

When comparing the daytime population–standardized DID values with the nighttime population–standardized values at the prefectural level, the former tended to be higher in bedroom communities such as Gifu, Nara, Kanagawa, Chiba, and Saitama, but lower in the central urban areas of Tokyo and Osaka. A similar trend was observed at the SMA level. These

**Table 1. Difference between nighttime population–standardized and daytime population–standardized defined daily doses per 1,000 population per day.**

| Rank | Prefecture/Secondary medical area | Difference |
|------|-----------------------------------|------------|
| Difference in AMU between nighttime population–standardized and daytime population–standardized (ranked in descending order) | | |
| 1 | Tokyo/Ku-chuoubu | 30.94 |
| 2 | Osaka/Osaka-shi | 4.94 |
| 3 | Tokyo/Ku-seibu | 4.50 |
| 4 | Tokyo/Ku-seinanbu | 3.15 |
| 5 | Fukushima/Soso | 2.30 |
| 6 | Aichi/Nagoya | 2.24 |
| 7 | Tokyo/Ku-nanbu | 1.99 |
| 8 | Fukuoka/Itoshima | 1.49 |
| 9 | Fukuoka/Noogata, Kurate | 1.34 |
| 10 | Aichi/Nishimikawa-hokubu | 1.25 |
| Difference in AMU between nighttime population–standardized and daytime population–standardized (ranked in ascending order) | | |
| 1 | Kyoto/Yamashiro-minami | -4.42 |
| 2 | Tokyo/Kitatama-hokubu | -3.78 |
| 3 | Fukuoka/Munakata | -3.70 |
| 4 | Kanagawa/Kawasaki-hokubu | -3.69 |
| 5 | Wakayama/Naga | -3.28 |
| 6 | Saitama/Nanbu | -2.91 |
| 7 | Aichi/Ama | -2.86 |
| 8 | Saitama/Keno | -2.85 |
| 9 | Nara/Seiwa | -2.82 |
| 10 | Chiba/Toukatsu-hokubu | -2.82 |
| Absolute value of the AMU difference between nighttime population–standardized and daytime population–standardized (ranked in ascending order) | | |
| 1 | Shimane/Masuda | 0.00 |
| 2 | Hokkaido/Tokachi | 0.00 |
| 3 | Yamagata/Murayama | 0.00 |
| 4 | Hokkaido/Kushiro | 0.00 |
| 5 | Gifu/Hida | 0.01 |
| 6 | Fukushima/Kennaka | 0.01 |
| 7 | Kouchi/Chuou | 0.01 |
| 8 | Kagoshima/Nansatsu | 0.01 |
| 9 | Kagoshima/Amami | 0.01 |
| 10 | Ehime/Matsuyama | 0.01 |

observations may be explained by the higher concentration of people in the city centers during the day for work or schooling (i.e., population outflow from bedroom communities during the day). In particular, there was a large difference between the DID values standardized for night-time population (43.08) and daytime population (12.15) in the Tokyo/Ku-chuoubu SMA, which experiences high population inflow during the day. However, substantial differences were mainly observed in the Kanto and Kansai regions that are centered around large metro-politan areas, with many other regions unaffected. An exception was the Fukushima/Soso region, which underwent evacuations due to the nuclear power plant disaster in 2011. The SMAs with small absolute differences tended to be located in rural regions and remote islands with higher levels of medical self-sufficiency (e.g., Hokkaido/Tokachi, Yamagata/Murayama,

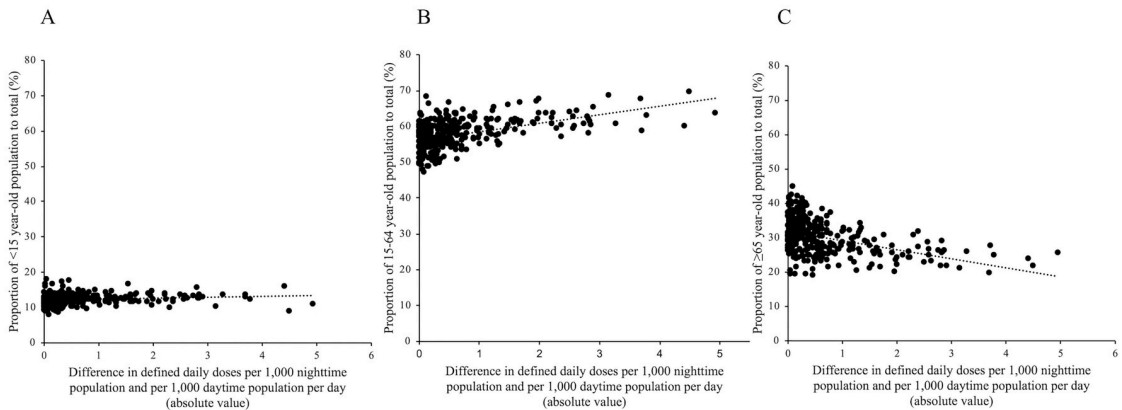

**Fig 4. Scatter plots of the proportion of age groups vs. absolute differences between DDDs per 1,000 nighttime population per day and DDDs per 1,000 daytime population per day.** (A) <15 years, (B) 15–64 years, and (C) ≥65 years. Each dot represents a secondary medical area. The correlation coefficients were 0.14 ($P = 0.0082$) for children (<15 years), 0.49 ($P < 0.001$) for working-age adults (15–64 years), and -0.44 ($P < 0.001$) for older adults (≥65 years). Data from Tokyo/Ku-chuoubu are not shown in the plots because this region was an extreme outlier (43.08).

Kagoshima/Amami, and Hokkaido/Kushiro), which would therefore have low population outflow.

Based on a hypothesis that population age structure may affect mobility across regional borders, we examined the correlations between the proportion of each age group in the population and the absolute difference in DID values. There was a significant and positive correlation between the proportion of working-age adults (who would be the most mobile among the age groups) and the absolute difference in DID values. Regions with a high proportion of working-age adults would experience higher population mobility, resulting in a larger difference in DID values between the different populations. In contrast, a significant and negative correlation was observed for older adults, who would have the least mobility among the age groups. Therefore, regions with many older adults would experience lower population mobility, resulting in a smaller difference in DID values between the different populations.

This study has several limitations. First, the NDB does not include information on the treatment of patients who pay their own expenses and those for whom the municipality shares the cost. However, almost all necessary medical services in Japan are covered by health insurance. Therefore, the NDB covers the majority of healthcare provided throughout the country. Next, daytime population statistics are only published once every five years in Japan. As these statistics are based on weekday estimates, weekends and holidays (accounting for approximately 30% of the year) are overlooked. Finally, the SMAs may differ from year to year due to the merging of municipalities. Although not examined in this study, AMU surveillance should consider such regional changes over time. Despite these limitations, our findings demonstrated the effects of population inflow and outflow on the population-standardized AMU.

Although regional AMU estimates would help to inform AMU-related policymaking, we recommend avoiding AMU evaluation in small regions standardized by the population number. If we wish to monitor AMU evaluations in small regions, only the temporal change in a region should be considered, for example.

## Conclusion

AMU surveillance has conventionally used the nighttime population for standardization. However, regional-level AMU estimates, especially of smaller regions such as SMAs, are more

susceptible to the influence of whatever population is used for standardization, which can lead to erroneous estimates. Therefore, new approaches are required to monitor AMU evaluations in small regions, for example, observing only the temporal changes in a region.

## Supporting information

**S1 Table. Prefectures, secondary medical areas, and municipalities in Japan.**
(DOCX)

**S2 Table. Daytime and nighttime populations at the national level, prefectural level, and secondary medical area level in Japan.**
(DOCX)

**S3 Table. Defined daily doses of antimicrobials at the national level, prefectural level, and secondary medical area level in Japan.**
(DOCX)

## Author Contributions

**Conceptualization:** Ryuji Koizumi, Yoshiki Kusama, Yuichi Muraki, Masahiro Ishikane, Daisuke Yamasaki, Masaki Tanabe.

**Data curation:** Ryuji Koizumi, Yoshiki Kusama, Yuichi Muraki, Masahiro Ishikane, Daisuke Yamasaki, Masaki Tanabe.

**Formal analysis:** Ryuji Koizumi.

**Funding acquisition:** Norio Ohmagari.

**Investigation:** Ryuji Koizumi.

**Methodology:** Ryuji Koizumi, Yoshiki Kusama, Yuichi Muraki, Masahiro Ishikane, Daisuke Yamasaki, Masaki Tanabe, Norio Ohmagari.

**Supervision:** Yoshiki Kusama, Yuichi Muraki, Masahiro Ishikane, Daisuke Yamasaki, Masaki Tanabe, Norio Ohmagari.

**Writing – original draft:** Ryuji Koizumi.

**Writing – review & editing:** Yoshiki Kusama, Yuichi Muraki, Masahiro Ishikane, Daisuke Yamasaki, Masaki Tanabe, Norio Ohmagari.

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
