## [Decision Letter · Decision Letter 0]

17 Nov 2020

PONE-D-20-15330

Effect of population inflow and outflow between rural and urban areas on regional antimicrobial use surveillance

PLOS ONE

Dear Dr. Kusama,

Thank you for submitting your manuscript to PLOS ONE. After careful consideration, we feel that it has merit but does not fully meet PLOS ONE’s publication criteria as it currently stands. Therefore, we invite you to submit a revised version of the manuscript that addresses the points raised during the review process.

Specifically, two experts in the field reviewed your manuscript. Their major concerns relate to statistical analyses and elaboration of the discussion.

We look forward to receiving your revised manuscript.

Kind regards,

Susan Hepp

Academic Editor

PLOS ONE

Journal Requirements:

2. In your ethics statement in the Methods section and in the online submission form, please clarify whether all data were fully anonymized before you accessed them.

3. Please state the date(s) when the data was obtained/accessed from the NDB.

4. Thank you for stating in your manuscript text

"This study was approved by the institutional review board of the National Center for Global Health and Medicine (Approval Number: NCGM-G-002505-00)."

Please also include this in your ethics statement in the online submission form.

5. Please include a table of relevant demographic details summarizing the studied population.

6. Please upload a copy of Figure 4, to which you refer in your text on page 16. If the figure is no longer to be included as part of the submission please remove all reference to it within the text.

7. Please include captions for your Supporting Information files at the end of your manuscript, and update any in-text citations to match accordingly. Please see our Supporting Information guidelines for more information: http://journals.plos.org/plosone/s/supporting-information

Reviewers' comments:

Reviewer's Responses to Questions

**Comments to the Author**

1. Is the manuscript technically sound, and do the data support the conclusions?

Reviewer #1: Yes

Reviewer #2: Partly

2. Has the statistical analysis been performed appropriately and rigorously? 

Reviewer #1: Yes

Reviewer #2: No

3. Have the authors made all data underlying the findings in their manuscript fully available?

Reviewer #1: Yes

Reviewer #2: Yes

4. Is the manuscript presented in an intelligible fashion and written in standard English?

Reviewer #1: Yes

Reviewer #2: Yes

5. Review Comments to the Author

Reviewer #1: Thank you for the opportunity to review this educational manuscript by Koizumi et al. They describe the effects of regional population inflow and outflow on estimates of antimicrobial use in Japan, and the implications this may have for future policies regarding antimicrobial stewardship. I have a few comments and questions that I hope will serve to improve the manuscript further:

1) The discussion is concise, but would benefit from a further elaboration of how this recommended shift from nighttime to daytime population usage will change antimicrobial use policy and management. What implications does this have for future regulation? What steps will be taken next? A more in-depth discussion would be helpful.

2) I did not receive Figure 4 – the description is provided, but not the figure itself. This should be provided and reviewed.

3) Figure 3 displays at a quite small size, making the color of the map and differences between region\\SMA difficult to see. Would enlarge.

Reviewer #2: Thank you for the opportunity to review this paper. This is an interesting manuscript presenting a results of a retrospective study that examine the impact of population inflow and outflow on sub- prefectural DID estimates in Japan.

My review mainly concerns only the statistical aspects of the study. Some questions reported below were raised and in my view, it is not acceptable in this form for the publication in this journal.

• The authors don’t’ compare median value of DID standardized between nighttime and daytime population. Are the difference statistically significant?

• In Figure 1 I think that the crosses represent the median and not the mean value as described in legend of figure 1.

• I suggest to the authors to estimate an indicator of agreement to evaluate the relationship between nighttime population–standardized DID values respect to daytime population–standardized DID values. (figure 2)

• Figure 4 is not included in the submission material

6. PLOS authors have the option to publish the peer review history of their article (what does this mean?). If published, this will include your full peer review and any attached files.

Reviewer #1: No

Reviewer #2: No

---

## [Author Response · Author response to Decision Letter 0]

14 Dec 2020

Response to Reviewers

Reviewer #1:

Response: Thank you for your time and effort in reviewing our manuscript. We have revised the manuscript in accordance with your suggestions. 

We have added a statistical analysis of the differences between the national-level DID values and the prefectural- and SMA-level DID values. This analysis showed that the SMA-level DID was significantly different from the national-level DID, irrespective of whether standardization was performed with the nighttime or daytime population. Therefore, as a major revision, we have softened the tone of our conclusion that we should use the daytime standardized–population as the SMA-level AMU’s denominator, and indicated that it may be better to avoid using SMA-level DID. Finally, we apologize for omitting Figure 4, and have included it in this resubmission. Thank you for giving us the chance to revise our manuscript, and we look forward to your evaluation.

Conclusions (Page 20, Lines 334-338)

Before

AMU surveillance has conventionally used the nighttime population for standardization. However, it is evident that nighttime population standardization can lead to erroneous estimates due to its omission of population inflow and outflow, especially for smaller regions. In Japan, the estimation of SMA-level DID values should be based on daytime population standardization to enable more impartial comparisons.

After

AMU surveillance has conventionally used the nighttime population for standardization. However, regional-level AMU estimates, especially of smaller regions such as SMAs, are more susceptible to the influence of whatever population is used for standardization, which can lead to erroneous estimates. Nevertheless, daytime populations that account for population inflow and outflow may be preferable to nighttime populations to minimize such errors and enable more precise comparisons.

1. The discussion is concise, but would benefit from a further elaboration of how this recommended shift from nighttime to daytime population usage will change antimicrobial use policy and management. What implications does this have for future regulation? What steps will be taken next? A more in-depth discussion would be helpful.

Thank you for your advice. As suggested, we have addressed the potential implications of our findings, although we prefer to refrain from a more in-depth discussion at this stage (we foresee that the shift from nighttime to daytime population could help produce more accurate AMU estimates, but anything downstream of that would be speculative). In this study, we identified population in/out flow to be one of the factors that can introduce bias into regional AMU estimates. As a next step, we aim to further elucidate the other factors that affect regional AMU, and to develop a standardization formula similar to the Standardized Antimicrobial Administration Ratio, which was developed by the US CDC and is used in hospital AMU comparisons. We have added this perspective to the end of the Discussion section.

Page 20, Lines 322-331

The shift from nighttime population to daytime population for standardization could provide more accurate regional-level AMU estimates, which would help to inform AMU-related policymaking, guide the efficient allocation of resources for region-specific antimicrobial stewardship programs, and contextualize the decision-making process for AMU regulations. As a next step, we aim to develop a formula to standardize regional-level AMU that can account for various sources of biases (e.g., population inflow and outflow) similar to the Standardized Antimicrobial Administration Ratio, which was developed by the US Centers for Disease Control and Prevention and is used in hospital-level AMU comparisons. In this way, our study represents the first step for developing a standardization method for regional-level AMU surveillance in Japan. Further studies are needed to identify the other factors of regional AMU surveillance.

Reference

14. van Santen K L, Edwards JR, Webb A K, Pollack L A, O’Leary E, Neuhauser M N, et al. The Standardized Antimicrobial Administration Ratio: A New Metric for Measuring and Comparing Antibiotic Use. Clin Infect Dis. 2018;67: 179-185. doi: 10.1093/cid/ciy075

2. I did not receive Figure 4 – the description is provided, but not the figure itself. This should be provided and reviewed.

We apologize for omitting Figure 4. This figure has been included in the resubmitted manuscript, and we would appreciate any comments or advice.

3. Figure 3 displays at a quite small size, making the color of the map and differences between region SMA difficult to see. Would enlarge.

Thank you for pointing this out. We have reformatted Figure 3 with a higher resolution (1200 dpi) and enlarged. We hope that it is more readable. 

Reviewer #2:

Response: Thank you for your time and effort in reviewing our manuscript. We have revised the manuscript in accordance with your suggestions. 

We have added a statistical analysis of the differences between the national-level DID values and the prefectural- and SMA-level DID values. This analysis showed that the SMA-level DID was significantly different from the national-level DID, irrespective of whether standardization was performed with the nighttime or daytime population. Therefore, as a major revision, we have softened the tone of our conclusion that we should use the daytime standardized–population as the SMA-level AMU’s denominator, and indicated that it may be better to avoid using SMA-level DID. Finally, we apologize for omitting Figure 4, and have included it in this resubmission. Thank you for giving us the chance to revise our manuscript, and we look forward to your evaluation.

Conclusions (Page 20, Lines 334-338)

Before

AMU surveillance has conventionally used the nighttime population for standardization. However, it is evident that nighttime population standardization can lead to erroneous estimates due to its omission of population inflow and outflow, especially for smaller regions. In Japan, the estimation of SMA-level DID values should be based on daytime population standardization to enable more impartial comparisons.

After

AMU surveillance has conventionally used the nighttime population for standardization. However, regional-level AMU estimates, especially of smaller regions such as SMAs, are more susceptible to the influence of whatever population is used for standardization, which can lead to erroneous estimates. Nevertheless, daytime populations that account for population inflow and outflow may be preferable to nighttime populations to minimize such errors and enable more precise comparisons.

1. The authors don’t’ compare median value of DID standardized between nighttime and daytime population. Are the differences statistically significant?

Thank you for your important advice. We have consulted with a statistician regarding this concern. Because prefectural- and SMA-level DID values do not have a “correct value” or gold standard, it would be difficult to interpret a statistically significant difference in median values between the nighttime and daytime populations. Instead, we have calculated the correlations between daytime and nighttime population–standardized DID in Figure 2. Furthermore, in Figure 1, we show the mean and 95% confidence interval for each indicator, and created violin plots to visualize their dispersions. Additionally, we compared each DID value to the national-level AMU using the one-sample t-test. The results showed that neither the daytime nor nighttime population–standardized DID at the prefectural level had any significant difference with the national-level AMU. However, both the daytime and nighttime population–standardized DID values at the SMA level were significantly different from the national-level AMU. Accordingly, we concluded that SMA-level DID values standardized using either of these populations would be difficult to evaluate. However, the 95% confidence interval was narrower and the mean (center of the widest section in the violin plot) was closer to the national AMU in the daytime population–standardized DID than in the nighttime population–standardized DID at the SMA level. Therefore, if SMA-level DID values must be calculated, the daytime population may represent the better option for standardization. We have added these points to the manuscript as follows:

Abstract (Page 3, Lines 38-42)

The national AMU was 17.21 DDDs per 1,000 population per day. The mean (95% confidence interval) prefectural-level DDDs per 1,000 nighttime and daytime population per day were 17.27 (14.10, 20.44) and 17.41 (14.30, 20.53), respectively. The mean (95% confidence interval) SMA-level DDDs per 1,000 nighttime and daytime population per day were 16.12 (9.84, 22.41) and 16.41 (10.57, 22.26), respectively.

Methods (Page 9, Lines 147 -150)

Next, we evaluated the distributions of daytime and nighttime population–standardized DID values at the prefectural and SMA levels using the Kolmogorov–Smirnov test. The mean DID values were compared with the national AMU using the one-sample t-test.

Results (Page 11, Lines 183-197)

When standardized with the nighttime population, the mean DID was 17.27 (95% confidence interval [14.10, 20.44]) at the prefectural level and 16.12 (95% confidence interval [9.84, 22.41]) at the SMA level. When standardized with the daytime population, the median DID was 17.41 (95% confidence interval [14.30, 20.53]) at the prefectural level and 16.41 (95% confidence interval [10.57, 22.26]) at the SMA level. Both the prefectural-level and SMA-level DID values were normally distributed regardless of the population used for standardization. The daytime and nighttime population–standardized mean DID values at the prefectural level were not significantly different from the national-level DID values (nighttime: P = .385; daytime: P = .811); however, the corresponding mean DID values at the SMA level were significantly different from the national-level DID values (nighttime: P < .001; daytime: P < .001). As shown in Fig 1, daytime population–standardized DID at the SMA level had a narrower dispersion and a mean value (center of the widest section in the violin plot) that was closer to the national-level DID than the nighttime population–standardized DID. In contrast, the daytime population and nighttime population–standardized DID at the prefectural level exhibited similar shapes in the violin plot.

Discussion (Page 17, Lines 269-279)

Previous studies have examined the effects of using different denominator values when calculating AMU in hospitals.9–12 However, these effects on regional AMU surveillance have not been explored. Even the World Health Organization’s AMU surveillance methodology does not address the appropriate methods for analyzing sub-national regions.13 In our analysis, SMA-level AMU (standardized using either the daytime or nighttime population) was found to be significantly different from the national-level AMU. This suggests that the population-standardized DID values of smaller regions are susceptible to the effects of population inflow and outflow, which can lead to erroneous results. 

At the SMA level, daytime population–standardization produced fewer outliers, narrower 95% confidence intervals, and mean DID values that were closer to the national-level DID than nighttime population–standardization.

Discussion (Page 18, Lines 283-284)

Although standardization using either population may be problematic for SMAs, our findings indicate that the use of the daytime population is preferable to the nighttime population for such adjustments.

2. In Figure 1 I think that the crosses represent the median and not the mean value as described in legend of figure 1.

Thank you for the suggestion. As mentioned in the response to your previous comment, we have replaced the boxplots with violin plots.

3. I suggest to the authors to estimate an indicator of agreement to evaluate the relationship between nighttime population–standardized DID values respect to daytime population–standardized DID values. (figure 2)

As advised, we calculated Pearson’s correlation coefficients and P values for Figure 2. The following sentences have been added:

Methods (Page 9, Lines 152-153)

Correlations between nighttime and daytime population–standardized DID values were examined using Pearson’s correlation coefficients.

Results (Page 12, Lines 205-207)

The correlation coefficient between nighttime and daytime population–standardized DID values was higher at the prefectural level (0.90; P < .001) than at the SMA level (0.80; P < .001).

4. Figure 4 is not included in the submission material

We apologize for omitting Figure 4. This figure has been included in the resubmitted manuscript, and we would appreciate any comments or advice.

---

## [Decision Letter · Decision Letter 1]

4 Feb 2021

PONE-D-20-15330R1

Effect of population inflow and outflow between rural and urban areas on regional antimicrobial use surveillance

PLOS ONE

Dear Dr. Kusama,

Thank you for submitting your manuscript to PLOS ONE. After careful consideration, we feel that it has merit but does not fully meet PLOS ONE’s publication criteria as it currently stands. Therefore, we invite you to submit a revised version of the manuscript that addresses the points raised during the review process.

We look forward to receiving your revised manuscript.

Kind regards,

Vijayaprakash Suppiah, PhD

Academic Editor

PLOS ONE

Reviewers' comments:

Reviewer's Responses to Questions

**Comments to the Author**

1. If the authors have adequately addressed your comments raised in a previous round of review and you feel that this manuscript is now acceptable for publication, you may indicate that here to bypass the “Comments to the Author” section, enter your conflict of interest statement in the “Confidential to Editor” section, and submit your "Accept" recommendation.

Reviewer #3: (No Response)

2. Is the manuscript technically sound, and do the data support the conclusions?

Reviewer #3: Partly

3. Has the statistical analysis been performed appropriately and rigorously? 

Reviewer #3: Yes

4. Have the authors made all data underlying the findings in their manuscript fully available?

Reviewer #3: No

5. Is the manuscript presented in an intelligible fashion and written in standard English?

Reviewer #3: Yes

6. Review Comments to the Author

Reviewer #3: Summary: Understanding rates of anti-microbial use is an important part of health system surveillance, however, these rates may be sensitive to the population size used in the denominator, specifically whether it represents the nighttime or daytime population. Here the authors calculate AMU rates standardied for both populations at the prefecture and secondary medical area across the country of Japan, finding that these two types of metrics differ, particularly for urban regions with high levels of daily commuting.

Strengths:

The analysis seems to be done correctly, and the writing is clear and succinct.

The authors did a very comprehensive job of responding to previous comments from reviewers, however I have a couple of small comments.

Figure 2 is very clear and does an excellent job conveying the results.

Major Comments:

The authors make claims about which population value (nighttime vs. daytime) should be used to calculate AMU, based on which is more correct. However, the datasets used include no information about the "true" AMU, so it is unclear to me what this claim in based on. The authors do rightly make claims as to where this difference in calculation could lead to the widest discrepancies between the two metrics, but I don't understand how this leads to the conclusion that it is better to use daytime populations. Perhaps there is a well-supported assumption that people are prescribed medicine near their work and school places, but I am not familiar with the Japanese healthcare system and so do not know if this is the case. Other readers may also not be familiar with Japanese healthcare, and it would be helpful if the authors could provide more reasoning for the case they make.

Minor Comments:

Line 162: Do you mean the populations size of each age category for each SMA?

Line 186: Why do you compare median here and mean on line 184? Is one of these a typo, since you do compare them later in this paragraph?

Table 1: What do the three different tables in Table 1 represent? Is it possible to add a sub-title, such as a row that seperates each table that lables what each one is?

Line 284: It is not clear to me why the daytime-population standardized DID is better (see comment above). Could you explain this further?

Fig. 3: Please label the color bar with what it represents.

Data Availability:

The authors have made the population data available in the supplement, but the data on AMU is not available, which is required to conduct the analyses.

7. PLOS authors have the option to publish the peer review history of their article (what does this mean?). If published, this will include your full peer review and any attached files.

Reviewer #3: No

---

## [Author Response · Author response to Decision Letter 1]

17 Feb 2021

Response to Reviewers

Reviewer #3:

Thank you for your useful advice. We have corrected our manuscript according to your suggestions. We hope that our revisions meet your requirements.

Major Comments:

The authors make claims about which population value (nighttime vs. daytime) should be used to calculate AMU, based on which is more correct. However, the datasets used include no information about the "true" AMU, so it is unclear to me what this claim in based on. The authors do rightly make claims as to where this difference in calculation could lead to the widest discrepancies between the two metrics, but I don't understand how this leads to the conclusion that it is better to use daytime populations. Perhaps there is a well-supported assumption that people are prescribed medicine near their work and school places, but I am not familiar with the Japanese healthcare system and so do not know if this is the case. Other readers may also not be familiar with Japanese healthcare, and it would be helpful if the authors could provide more reasoning for the case they make.

Thank you for pointing this out. We agree with your opinion. Although we described that daytime population standardization is a superior method for estimating AMU in small regions than nighttime population standardization, this is not justified because of the absence of a standard AMU, as you pointed out. Therefore, we have corrected the Discussion and Conclusions as follows: 

Abstract (Lines 47-49)

Before

Regional-level AMU estimates, especially of smaller regions such as SMAs, are susceptible to the use of different populations for standardization. Daytime populations that account for population in/outflow may be preferable to nighttime populations for such adjustments.

After

Regional-level AMU estimates, especially of smaller regions such as SMAs, are susceptible to the use of different populations for standardization. This finding indicates that AMU standardization based on population values is not suitable for AMU estimates in small regions. 

Discussion (Line 277-284)

Before

At the SMA level, daytime population–standardization produced fewer outliers, narrower 95% confidence intervals, and mean DID values that were closer to the national-level DID than nighttime population–standardization. These findings showed that when analyzing smaller regional units such as SMAs, the use of different populations for standardization had a considerable effect on DID estimates in central urban areas and their surrounding regions. As more medical examinations and prescriptions are received during the day than at night, the calculation of regional AMU indices should account for the effects of population inflow and outflow. Although standardization using either population may be problematic for SMAs, our findings indicate that the use of the daytime population is preferable to the nighttime population for such adjustments.

After

As more medical examinations and prescriptions are received during the day than at night, the calculation of regional AMU indices should account for the effects of population inflow and outflow. At the SMA level, daytime population standardization produced fewer outliers, narrower 95% confidence intervals, and mean DID values that were closer to the national DID than nighttime population standardization. These findings showed that when analyzing smaller regional units, such as SMAs, the use of different populations for standardization has a considerable effect on DID estimates in central urban areas and their surrounding regions. Our insights indicate that AMU standardization based on population values is not suitable for AMU estimates in small regions.

Discussion (Line 319-322)

Before

The shift from nighttime population to daytime population for standardization could provide more accurate regional-level AMU estimates, which would help to inform AMU-related policymaking, guide the efficient allocation of resources for region-specific antimicrobial stewardship programs, and contextualize the decision-making process for AMU regulations. As a next step, we aim to develop a formula to standardize regional-level AMU that can account for various sources of biases (e.g., population inflow and outflow) similar to the Standardized Antimicrobial Administration Ratio, which was developed by the US Centers for Disease Control and Prevention and is used in hospital-level AMU comparisons. In this way, our study represents the first step for developing a standardization method for regional-level AMU surveillance in Japan. Further studies are needed to identify the other factors of regional AMU surveillance.

After

Although regional AMU estimates would help inform AMU-related policymaking, we recommend avoiding AMU evaluation in small regions standardized by the population number. If we wish to monitor AMU evaluations in small regions, only the temporal change in a region should be considered, for example. 

Conclusion (Line 328-329)

Before

Nevertheless, daytime populations that account for population inflow and outflow may be preferable to nighttime populations to minimize such errors and enable more precise comparisons.

After

Therefore, new approaches are required to monitor AMU evaluations in small regions, for example, observing only the temporal changes in a region.

Minor Comments:

1. Line 162: Do you mean the populations size of each age category for each SMA?

Yes, we used population numbers according to the age categories. We have corrected the description as you indicated.

Methods (Line 160-161)

Before

Finally, we divided each SMA’s population into three categories based on age (children: <15 years, working-age adults: 15–64 years, and older adults: ≥65 years), and analyzed the correlation between the absolute difference in DID values and each age category. For this analysis, the Tokyo/Ku-chuoubu SMA was again excluded as it was an extreme outlier.

After

Finally, we used the population number for each age category (children: <15 years, working-age adults: 15–64 years, and older adults: ≥65 years) for each SMA, and analyzed the correlation between the absolute difference in DID values and each age category. For this analysis, the Tokyo/Ku-chuoubu SMA was excluded as it was an extreme outlier.

2. Line 186: Why do you compare median here and mean on line 184? Is one of these a typo, since you do compare them later in this paragraph?

Thank you for pointing this out. We corrected “median” to “mean.”

3. Table 1: What do the three different tables in Table 1 represent? Is it possible to add a sub-title, such as a row that seperates each table that lables what each one is?

We added descriptions to each table. 

4. Line 284: It is not clear to me why the daytime-population standardized DID is better (see comment above). Could you explain this further?

Thank you. We corrected our discussion and conclusions as presented above.

5. Fig. 3: Please label the color bar with what it represents.

Thank you. We have added explanations of the color bars. 

6. The authors have made the population data available in the supplement, but the data on AMU is not available, which is required to conduct the analyses.

Thank you. We have created Supplementary Table 3 to display AMU data.

Methods (Line 155-156)

each DID value was shown in Table S3

---

## [Decision Letter · Decision Letter 2]

25 Feb 2021

Effect of population inflow and outflow between rural and urban areas on regional antimicrobial use surveillance

PONE-D-20-15330R2

Dear Dr. Kusama,

We’re pleased to inform you that your manuscript has been judged scientifically suitable for publication and will be formally accepted for publication once it meets all outstanding technical requirements.

Kind regards,

Vijayaprakash Suppiah, PhD

Academic Editor

PLOS ONE

Reviewers' comments:

Reviewer's Responses to Questions

**Comments to the Author**

1. If the authors have adequately addressed your comments raised in a previous round of review and you feel that this manuscript is now acceptable for publication, you may indicate that here to bypass the “Comments to the Author” section, enter your conflict of interest statement in the “Confidential to Editor” section, and submit your "Accept" recommendation.

Reviewer #3: All comments have been addressed

2. Is the manuscript technically sound, and do the data support the conclusions?

Reviewer #3: Yes

3. Has the statistical analysis been performed appropriately and rigorously? 

Reviewer #3: Yes

4. Have the authors made all data underlying the findings in their manuscript fully available?

Reviewer #3: Yes

5. Is the manuscript presented in an intelligible fashion and written in standard English?

Reviewer #3: Yes

6. Review Comments to the Author

Reviewer #3: The authors have addressed all of my comments. I find the new paragraph in the discussion regarding how daytime population numbers affect the AMU indices especially helpful and I think it really ties the manuscript together. Great work!

7. PLOS authors have the option to publish the peer review history of their article (what does this mean?). If published, this will include your full peer review and any attached files.

Reviewer #3: No

---

## [Editor Report · Acceptance letter]

2 Mar 2021

PONE-D-20-15330R2 

Effect of population inflow and outflow between rural and urban areas on regional antimicrobial use surveillance 

Dear Dr. Kusama:

I'm pleased to inform you that your manuscript has been deemed suitable for publication in PLOS ONE. Congratulations! Your manuscript is now with our production department. 

Kind regards, 

on behalf of

Dr. Vijayaprakash Suppiah 

Academic Editor

PLOS ONE